# Impact of COVID-19 on Antimicrobial Consumption and Spread of Multidrug-Resistance in Bacterial Infections

**DOI:** 10.3390/antibiotics11040535

**Published:** 2022-04-18

**Authors:** Kibum Jeon, Seri Jeong, Nuri Lee, Min-Jeong Park, Wonkeun Song, Han-Sung Kim, Hyun Soo Kim, Jae-Seok Kim

**Affiliations:** 1Department of Laboratory Medicine, Hallym University Hangang Sacred Heart Hospital, Seoul 07247, Korea; pourmythe@naver.com; 2Department of Laboratory Medicine, Hallym University Kangnam Sacred Heart Hospital, Seoul 07442, Korea; nurilee822@hallym.or.kr (N.L.); mjpark@hallym.or.kr (M.-J.P.); swonkeun@naver.com (W.S.); 3Department of Laboratory Medicine, Hallym University Sacred Heart Hospital, Anyang 14068, Korea; kimhs@hallym.or.kr; 4Department of Laboratory Medicine, Hallym University Dongtan Sacred Heart Hospital, Hwaseong 18450, Korea; kimhyun@hallym.or.kr; 5Department of Laboratory Medicine, Kangdong Sacred Heart Hospital, Seoul 05355, Korea; jaeseok@kdh.or.kr

**Keywords:** COVID-19, pandemic, antibiotic, antimicrobial resistance, multidrug resistance, prevalence

## Abstract

The spread of COVID-19 pandemic may have affected antibiotic consumption patterns and the prevalence of colonized or infected by multidrug-resistant (MDR) bacteria. We investigated the differences in the consumption of antibiotics easily prone to resistance and the prevalence of MDR bacteria during the COVID-19 pandemic (March 2020 to September 2021) compared to in the pre-pandemic period (March 2018 to September 2019). Data on usage of antibiotics and infections caused by methicillin-resistant *Staphylococcus aureus* (MRSA), vancomycin-resistant *Enterococcus* (VRE), carbapenem-resistant *Enterobacteriaceae* (CRE), carbapenem-resistant *Acinetobacter baumannii* (CRAB), and carbapenem-resistant *Pseudomonas aeruginosa* (CRPA) were obtained from hospitalized patients in four university hospitals. The consumption of penicillin with β-lactamase inhibitors (3.4% in ward, 5.8% in intensive care unit (ICU)), and carbapenems (25.9% in ward, 12.1% in ICU) increased during the pandemic period. The prevalence of MRSA (4.7%), VRE (49.0%), CRE (22.4%), and CRPA (20.1%) isolated in clinical samples from the ward and VRE (26.7%) and CRE (36.4%) isolated in clinical samples from the ICU were significantly increased, respectively. Meanwhile, only the prevalence of CRE (38.7%) isolated in surveillance samples from the ward increased. The COVID-19 pandemic is associated with increased consumption of antibiotics and has influenced the prevalence of infections caused by MDR isolates.

## 1. Introduction

Coronavirus disease 2019 (COVID-19) has spread worldwide since it was first identified in December 2019 in China. This pandemic, which has been rampant for over two years, has caused 6,170,283 cumulative deaths up to 11 April 2022, worldwide [1] and has a huge impact on social and healthcare systems. South Korea reported its first COVID-19 case in January 2020, and so far, 15,424,598 cumulative confirmed cases and 19,679 deaths were reported [2].

This COVID-19 pandemic has brought about many changes in disease prevention protocols, including wearing a mask, washing hands, and social distancing. It has also affected hospitals’ guidelines, including COVID-19 screening tests and isolation at admission, wearing of reinforced protective equipment, and the expansion of negative pressure wards [3]. The lifestyle changes in the general population and strict management of COVID-19 in the hospitals have reduced the spread of several respiratory infectious diseases [4,5,6]. However, increased susceptibility to infection precautions can lead to the overuse of antibiotics in hospitalized patients, resulting in the increased antimicrobial resistance [7]. Infections caused by multidrug-resistant (MDR) bacteria have a strong correlation with in-hospital mortality, especially in critically ill patients. Among them, methicillin-resistant *Staphylococcus aureus* (MRSA), vancomycin-resistant *Enterococcus* (VRE), carbapenem-resistant *Enterobacteriaceae* (CRE), carbapenem-resistant *Acinetobacter baumannii* (CRAB), and carbapenem-resistant *Pseudomonas aeruginosa* (CRPA) have been a huge challenge in control of healthcare-associated infection for a few decades [8,9,10,11,12,13]. Since the excessive use of antibiotics plays an important role in the increase of MDR bacteria, prevention of antibiotic overuse is one of the important infection control measures [14,15,16,17,18]. The overuse of broad-spectrum antibiotics, a concern in the era of the COVID-19 pandemic, cannot be ignored, as it can affect the prevalence of MDR bacteria. In addition, medical competence in the pandemic period is concentrated on COVID-19 [19,20]; therefore, management of MDR bacteria may be somewhat loosened, which may further affect the prevalence of MDR bacteria.

There have been several review reports for the differences in antibiotic use and prevalence of MDR bacteria before and during the COVID-19 pandemic. The impact of COVID-19 on AMR varied depending on the health care environment and public health policies of each country [21]. Studies of changes in antibiotic consumption and antibiotic resistance in individual countries during the COVID-19 period are important for developing strategies to overcome antimicrobial resistance [22]. Experts suggested that this pandemic is an opportunity to accelerate the implementation of a One Health approach to tackle the AMR crisis at the global scale [23]. In addition, stratified approach of the differences in antibiotic use patterns in the intensive care unit (ICU) and general wards, and the prevalence of MDR bacteria between clinical and surveillance samples are scarcely reported. Therefore, this study investigated the effect of the COVID-19 pandemic on the use of representative antibiotics, including penicillin with β-lactamase inhibitors, vancomycin, and carbapenems, and the prevalence of MDR bacteria by location (ICU and ward) and sample type (clinical and surveillance) of four university hospitals in South Korea compared to pre-pandemic era. Among MDR-bacteria, MRSA, VRE, CRE, CRAB, and CRPA are designated as legal communicable disease in south Korea and when detected in a patient’s samples, it must be reported to the Korea Centers for Disease Control and Prevention (KCDC) and are strictly managed. Clinical data warehouse from multi university hospitals in different regions were available for the prevalence of these five MDR bacteria and associated antibiotic usages. These enables including a large population in this study.

## 2. Results

### 2.1. Comparison of Consumption of Antibiotic Agents

Total consumption of penicillin with β-lactamase inhibitors was 72.48 defined daily dose (DDDs)/1000 patient-days and 115.78 DDDs/1000 patient-days in the wards and ICU before the pandemic, respectively. During the pandemic, 74.97 DDDs/1000 patient-days and 122.53 DDDs/1000 patient-days were prescribed in the wards and ICU, respectively. A 3.4% increase in wards and 5.8% increase in ICU was observed (*p* < 0.001). Vancomycin consumption in the ward increased by 16.7% from 11.58 DDDs/1000 patient-days in the pre-pandemic period to 13.52 DDDs/1000 patient-days during the pandemic (*p* < 0.001). Vancomycin consumption in the ICU decreased by 4% from 49.35 DDDs/1000 patient-days in the pre-pandemic period to 47.37 DDDs/1000 patient-days during the pandemic (*p* < 0.001), conversely. Total consumption of carbapenem in the wards increased by 25.9% from 30.15 DDDs/1000 patient-days in the pre-pandemic period to 37.96 DDDs/1000 patient-days in pandemic period and increased by 12.1% in the ICU from 123.99 DDDs/1000 patient-days days in the pre-pandemic period to 139.0 DDDs/1000 patient-days in pandemic period, showing the largest increase in the consumption of included antibiotics in both locations (Table 1).

### 2.2. Prevalence of Infection Caused by MDR Bacteria

#### 2.2.1. Methicillin-Resistant Staphylococcus Aureus

The prevalence of MRSA in clinical samples was 0.86 and 6.27 infection cases/1000 patient-days from the wards and ICU before the pandemic, respectively. During the pandemic, the prevalence of MRSA in the wards increased to 0.90 infection cases/1000 patient-days (+4.7%, *p* < 0.001); however, prevalence in ICU decreased to 4.30 infection cases/1000 patient-days (−31.4%, *p* < 0.001; Table 2). According to the linear regression analysis, MRSA tended to decrease before the pandemic in the ICU clinical samples (r^2^ = 0.810, *p* = 0.006). However, this decrease was alleviated during the pandemic period (Figure 1).

In surveillance samples, the prevalence of MRSA in the wards showed the lowest prevalence (0.04 infection cases/1000 patient-days) among those with MDR bacteria isolates and showed no significant change during the pandemic compared to the pre-pandemic period. Notably, the prevalence of MRSA in the ICU significantly decreased from 6.34 infection cases/1000 patient-days in pre-pandemic period to 3.87 infection cases/1000 patient-days during the pandemic period (−38.9%, *p* < 0.001; Table 3).

#### 2.2.2. Vancomycin-Resistant Enterococcus

The prevalence of VRE in clinical samples was 0.46 and 1.51 infection cases/1000 patient-days in the wards and ICU before the pandemic, respectively. It increased significantly during the pandemic in both location (ward: 49%; ICU: 26.7%, *p* < 0.001). In surveillance samples, the prevalence of VRE in the ICU showed 3.76 infection cases/1000 patient-days in pre-pandemic period and 1.31 infection cases/1000 patient-days during pandemic, showed decreased significantly (−65.2%, *p* < 0.001), whereas that in the ward showed subtle changes (−8.7%, *p* = 0.475). Linear analyses revealed that VRE decreased significantly during the pandemic in surveillance samples in the ICU compared to the pre-pandemic period (r^2^ = 0.734, *p* = 0.029; Figure 2).

The prevalence of infection cases caused by *Enterococcus faecium* rather than *Enterococcus faecalis* was dominant regardless of the sample type and location. *Enterococcus faecium* increased significantly in clinical samples from ward (+50.1%, *p* < 0.001) and decreased in the surveillance samples from the ICU (−65.1%, *p* < 0.001) during pandemic. *Enterococcus faecalis* showed no significant changes during the pandemic in both location and sample (Appendix A).

#### 2.2.3. Carbapenem-Resistant Enterobacteriaceae

The prevalence of CRE in clinical samples was 0.23 and 1.03 infection cases/1000 patient-days in the wards and ICU before the pandemic, respectively. These prevalences in clinical samples increased significantly during the pandemic (ward: 22.4%; ICU: 36.4%, *p* < 0.001; Table 2). In surveillance samples, prevalence from the wards increased from 0.52 infection cases/1000 patient-days in pre-pandemic period to 0.73 infection cases/1000 patient-days in pandemic period (38.7%, *p* < 0.001); however, no significant changes were observed with the samples from the ICU (Table 3). Of note, CRE significantly increased during the pandemic in surveillance samples from the wards (r^2^ = 0.725, *p* = 0.031; Figure 2a). Although CRE in the ICU did not show a significant difference, linear analysis revealed an increase during the pandemic (r^2^ = 0.751, *p* = 0.025; Figure 2b).

*Klebsiella pneumoniae* was dominant in both clinical and surveillance samples regardless of location. *K. pneumoniae* increased significantly in both sample type and location during the pandemic (45.4% for clinical samples in ward, 67.6% for clinical samples in ICU, 52.4% for surveillance sample in ward, *p* = 0.001, and 10.7% for surveillance samples in ICU, *p* = 0.009; Appendix A). *Escherichia coli* also showed a significant increase in the wards during the pandemic (58.0% for clinical samples and 45.2% for surveillance samples). In surveillance samples, *Enterobacter* species decreased significantly in the ICU (38.0%, *p* = 0.016) and other *Klebsiella* species such as *K. aerogenes*, *K. oxytoca*, *K. ozaenae*, and *K. variicola* increased in the ward during the pandemic (85.7%, *p* = 0.010).

#### 2.2.4. Carbapenem-Resistant Acinetobacter Baumannii

There was no significant change in the prevalence of CRAB in the clinical and surveillance samples in the ward. However, the prevalence of CRAB in the ICU revealed a decrease in both sample types during the pandemic (−18.6% for clinical samples, *p* = 0.003, and −24.6% for surveillance samples, *p* = 0.012; Table 2 and Table 3). Based on the trend analysis, the decreasing slopes in clinical samples in the wards and ICU during the pre-pandemic period were lower during the pandemic (Figure 1).

#### 2.2.5. Carbapenem-Resistant Pseudomonas Aeruginosa

Prevalence of CRPA increased significantly in clinical samples from the ward during the pandemic (20.1%, *p* < 0.001), whereas it decreased in the ICU (−25.7%, *p* = 0.005). Surveillance culture of CRPA was conducted in only two hospitals (C and D), and CRPA was not detected in both locations during the entire study period.

## 3. Discussion

### 3.1. Consumption of Antibiotic Agents in Pandemic

According to our study, the overall consumption of antibiotics increased among inpatients during the COVID-19 pandemic (Table 1). However, there are some differences among hospitals. The difference seems to depend on the size of each hospital and the characteristics of the hospitalized patients. Carbapenems were most frequently used in the ICU, and their consumption significantly increased in the wards (25.9%) and ICU (12.1%). Consistent with our study, the consumption of carbapenems at National Taiwan University Hospital increased by 13.8% from 330.4 to 376.0 DDDs/1000 patient-days [24]. Antimicrobial stewardship programs to minimize the use of empirical antibiotics may be somewhat loosened because of immense burden of COVID-19. The use of penicillin with β-lactamase inhibitors increased in wards (3.4%) and in the ICU (5.8%), and a similar increase (4.1%) in use of β-Lactam/β-lactamase inhibitor combinations was observed in a previous study [24]. In contrast, Ryu et al. [25] reported a reduction in the consumption of penicillin with β-lactamase inhibitors during the COVID-19 pandemic in South Korea. Another study for assessing the impact of the COVID-19 pandemic in Spanish hospital showed that antibiotic use increased for six weeks after the national lockdown, followed by a sustained reduction [26]. The length of the study period, size of the population assessed, and type of antibiotics investigated could result in these differences. We included the recent pandemic period until September 2021, which was more than a year longer than that of a previous study [25]. In terms of the study population, we focused on hospitalized patients rather than outpatients. In addition, the compositions of the antimicrobial agents were different.

### 3.2. Prevalence of Infection Caused by MRSA

The prevalence of MRSA increased in clinical samples from the ward (4.7%) (Table 2). An increase in MRSA was observed in a previous study on COVID-19 and antimicrobial resistance in Greece [27], consistent with our data. A higher incidence of infection caused by MRSA in patients with COVID-19 has been reported in several studies [28,29]. Additionally, analysis of the lung microbiota indicated that *Staphylococcus aureus* was enriched in these patients [29]. Although MRSA decreased in the ICU (−31.4%), MRSA tended to decrease before the pandemic and was alleviated during the pandemic based on regression analysis, concordant with the increased consumption of penicillin with β-lactamase inhibitors.

### 3.3. Prevalence of Infection Caused by VRE

The prevalence of VRE substantially increased, especially in clinical samples from both location (ward, 49% and ICU, 26.7%; Table 2 and Appendix A). Among VRE, the increase in *Enterococcus faecium* in clinical samples was also prominent in a previous study in Greece [27]. In contrast to clinical samples, surveillance samples showed a decreased prevalence (Table 3). Trend analysis also showed a decreasing slope in surveillance samples in the ICU. These findings support close monitoring and appropriate interventions to prevent the spread of VRE, particularly in clinical samples.

### 3.4. Prevalence of Infection Caused by CRE 

Based on our data, CRE increased during the pandemic in clinical samples from both location (ward, 22.4% and ICU, 36.4%) (Table 2) and in surveillance sample from ward (38.7%) (Table 3). Although surveillance samples in the ICU did not show significant changes, trend analysis revealed an increasing trend, consistent with the increasing consumption of carbapenems. Tiri et al. [30] reported that the incidence of CRE increased from 6.7% in 2019 to 50% in March–April 2020. The most commonly isolated species among CRE was *K. pneumoniae*, which is consistent with our results (Appendix A). The increase in antimicrobial resistance in ICUs due to the spread of high-risk clones of CRE during the COVID-19 pandemic has also been reported [31,32]. The breakdown of antimicrobial stewardship and infection control programs determine an increase in infections caused by MDR bacteria [32]. Antimicrobial stewardship programs to minimize the use of empirical antibiotics during the COVID-19 pandemic are required to reduce the burden of healthcare-associated infections [33].

### 3.5. Prevalence of Infection Caused by CRAB

In contrast to CRE, CRAB showed non-significant changes in ward and decrease in ICU (−18.6% for clinical samples and −24.6% for surveillance samples) (Table 2 and Table 3). However, the decreasing trend derived from infection control for MDR bacteria slowed based on trend analysis. Immense COVID-19 burden on healthcare workers might affect infection control activities, such as monitoring hygiene practices, leading to the slowdown of CRAB [24]. In another study, CRAB isolates increased with changes from decreasing to increasing trends [27]. The increased clinical severity and duration of hospitalization [34], as well as increased consumption of carbapenems, would cause these patterns.

### 3.6. Prevalence of Infection Caused by CRPA

The prevalence of CRPA varied (20.1% in the ward and −25.7% in the ICU) (Table 2). In previous studies, high resistance to antibiotics and coinfection with COVID-19 patients causing worse outcomes of CRPA have been reported [35,36]. Stratified analysis and control for the ward rather than the ICU are necessary for CRPA based on our data.

In our study, we observed the use of antibiotics in the COVID-19 pandemic has increased and influenced on the prevalence of infections caused by MDR isolates. In particular, prevalence of MDR infection, which had been decreased before the pandemic, gradually increased toward the third quarter of 2021 (Figure 1 and Figure 2). This may be due to an excessively prolonged pandemic has increased the fatigue of medical capacity, leading to the overuse of antibiotics. Ansari et al. [7] also reported COVID-19 pandemic may contribute to worsening the scope of antimicrobial resistance globally through the non-rational use of antibiotics as part of preventive and therapeutic management of COVID-19.

Our study was unique in the following aspects: (i) The recent COVID-19 pandemic period of more than one year was included in the study. (ii) Multiple tertiary hospitals in different regions of Korea were included. (iii) To the best of our knowledge, this is the first study to stratify clinical and surveillance samples for the changes in the prevalence of MDR bacteria during COVID-19 in South Korea. (iv) A concurrent investigation of antibiotic usage and the prevalence of infection was performed, and trend analysis for their association was also conducted.

Although including multiple hospitals with diverse samples is the advantage of this study, different commercial platforms used for identification and antimicrobial susceptibility testing might influence the results. Further studies including more hospitals with a larger study population would be helpful to gain insight into the trends of COVID-19.

## 4. Materials and Methods

### 4.1. Study Design

Data on the use of antibiotics prone to resistance and the prevalence of resistance in hospitalized patients were collected from Clinical Data Warehouse, which included clinical information from four university hospitals in South Korea (Hallym University Hangang Sacred Heart Hospital with 158 beds [A] and Hallym University Kangnam Sacred Heart Hospital with 572 beds [B] in Seoul, and Hallym University Dongtan Sacred Heart Hospital with 660 beds [C] and Hallym University Sacred Heart Hospital with 834 beds [D] in Gyeonggi-do), between March 2018 and September 2021. The collected data included the hospitalization period, antibiotic dosage, microbial culture results (species of isolated bacteria), and antimicrobial susceptibility test results. A total of 209,107 hospitalized patients were included: 8208, 54,130, 75,353, and 71,416 from hospitals A, B, C, and D, respectively. The included data were stratified according to location of isolation (wards or ICUs). The study periods were split into two: the “pre-pandemic” period from March 2018 to September 2019 and the “pandemic” period from March 2020 to September 2021. In South Korea, the national public health alert was raised to its highest level near March 2020. Differences in antibiotic consumption and antimicrobial resistance between the pre-pandemic and the pandemic periods were assessed.

### 4.2. Antibiotic Consumption

Consumption of representative antibiotics, such as penicillin with β-lactamase inhibitors (amoxicillin-clavulanate, ampicillin-sulbactam, and piperacillin-tazobactam), vancomycin, and carbapenems (imipenem, meropenem, doripenem, and ertapenem) was analyzed. The defined daily dose (DDD) was utilized to estimate the antibiotic consumption of the study population [37]. The DDD of the included antibiotics was analyzed according to the World Health Organization Collaborating Centre for Drug Statistics Methodology [38]. The DDDs used for each drug consumption were as follows: (i) 1.5 g, 3 g, 6 g, and 14 g of oral penicillin, and parenteral amoxicillin-clavulanate, ampicillin-sulbactam, and piperacillin-tazobactam, respectively; (ii) 2 g, 3 g, 1.5 g, and 1 g of imipenem, meropenem, doripenem, and ertapenem; (iii) 2 g of vancomycin.

### 4.3. Identification of MDR Bacteria and Antimicrobial Susceptibility Testing

Microbial identification and antimicrobial susceptibility tests were performed on clinical samples taken during the work-up of patients with suspected infections and surveillance culture samples for epidemiological and infection control purposes from hospitalized patients. Bacteria were identified using a Vitek 2 system (bioMérieux, Hazelwood, MO, USA) at hospitals A and C, a matrix-assisted laser desorption ionization-time-of-flight mass spectrometry on a Vitek MS instrument (bioMérieux) at hospital B, and a MicroScan Walkaway-96 system (Siemens, West Sacramento, CA, USA) at hospital D. The minimal inhibitory concentrations (MICs) of isolates were measured using the Vitek 2 system (bioMérieux) at hospitals A, B, and C, and the MicroScan Walkaway-96 system at hospital D. MRSA, VRE, CRE, CRAB, and CRPA were defined as bacterial species with MICs > 4 μg/mL for oxacillin, >8 μg/mL for vancomycin, and >0.5 μg/mL for ertapenem or MIC > 1 μg/mL for imipenem or meropenem. MIC breakpoints were applied according to the Clinical and Laboratory Standards Institute guidelines (M100S) [39]. In total, 4864 MRSA, 3361 VRE, 1604 CRE, 5320 CRAB, and 1787 CRPA strains were isolated from hospitalized patients. The first isolate of a given species recovered during the admission period for each patient was included.

### 4.4. Statistical Analyses

Antibiotic consumption was presented as DDDs/1000 patient-days. The prevalence of non-susceptibility was calculated by dividing the number of colonized or infected patients by 1000 patient-admission days. The Mann–Whitney test was applied to compare nonparametric quantitative variables between pre-pandemic and pandemic periods, and either the Pearson’s chi-square test or Fisher’s exact test was used to compare categorical variables. Linear regression analysis was used to assess the trend of each prevalence per quarter. Statistical analyses were performed using SPSS Statistics software version 24 (IBM Corp. Released 2018. IBM SPSS Statistics for Windows, Version 24.0. Armonk, NY, USA: IBM Corp.). For all statistical tests used in the study, *p* values < 0.05 were considered statistically significant. All data that were included in the analysis were previously anonymized.

## 5. Conclusions

The COVID-19 pandemic has had a major impact on the use of antibiotics and the prevalence of MDR bacteria. Antibiotic consumption, particularly consumption of penicillin with β-lactamase inhibitors and carbapenems, increased during the pandemic period. The prevalence of MRSA, VRE, CRE, and CRPA in the ward and VRE and CRE in the ICU from clinical samples increased. For surveillance samples, the prevalence of CRE in the ward increased. The slowdown of the decreasing trend of MDR bacteria was observed through trend analysis. Maintaining infection control protocols stratified by sample type and location as well as minimization of empirical antibiotic use during long-lasting COVID-19 pandemic should be maintained to prevent the spread of infection by MDR isolates.

## Figures and Tables

**Figure 1 antibiotics-11-00535-f001:**
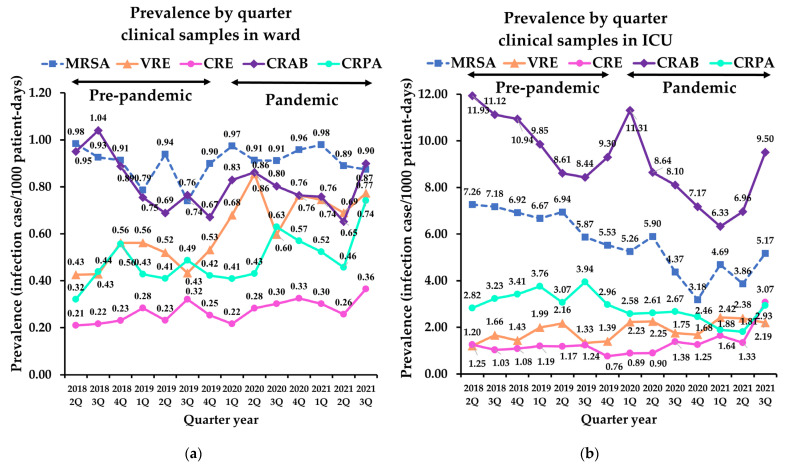
Quarterly prevalence of multidrug-resistant bacteria isolates from clinical samples: (**a**) prevalence in word; (**b**) prevalence in ICU. MRSA, methicillin-resistant *Staphylococcus aureus*; VRE, vancomycin-resistant *Enterococcus*; CRE, carbapenem-resistant *Enterobacteriaceae*; CRAB, carbapenem-resistant *Acinetobacter baumannii*; CRPA, carbapenem-resistant *Pseudomonas aeruginosa*; ICU, intensive care unit.

**Figure 2 antibiotics-11-00535-f002:**
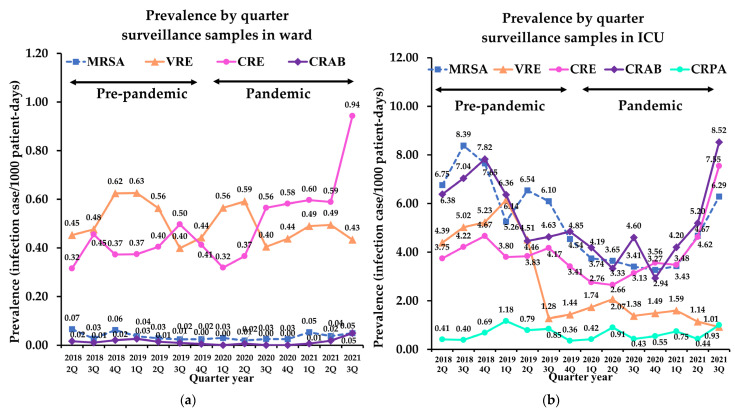
Quarterly prevalence of multidrug-resistant bacteria isolates from surveillance samples: (**a**) prevalence in wards; (**b**) prevalence in ICUs. MRSA, methicillin-resistant *Staphylococcus aureus*; VRE, vancomycin-resistant *Enterococcus*; CRE, carbapenem-resistant *Enterobacteriaceae*; CRAB, carbapenem-resistant *Acinetobacter baumannii*; CRPA, carbapenem-resistant *Pseudomonas aeruginosa*; ICU, intensive care unit.

**Table 1 antibiotics-11-00535-t001:** Antibiotic consumption during indicated period in hospitalized patients.

		Ward				ICU			
Antibiotics	Hospital	March 2018 –September 2019	March 2020 –September 2021	% Change	*p*	March 2018 –September 2019	March 2020 –September 2021	% Change	*p*
**Penicillin** **with β-lactamase inhibitors**	Total	72.48	74.97	+3.4	**<0.001**	115.78	122.53	+5.8	**<0.001**
	A	69.60	106.17	+52.5	**<0.001**	195.13	258.44	+32.4	**<0.001**
	B	32.07	28.65	−10.7	**<0.001**	23.44	22.79	−2.8	**<0.001**
	C	75.37	72.71	−3.5	0.385	124.60	125.42	+0.7	0.847
	D	95.91	100.43	+4.7	**<0.001**	170.19	185.46	+9.0	**<0.001**
**Vancomycin**	Total	11.58	13.52	+16.7	**<0.001**	49.35	47.37	−4.0	**<0.001**
	A	6.01	9.48	+57.8	**<0.001**	86.82	86.80	−0.0	**0.024**
	B	9.62	8.67	−9.9	**<0.001**	33.16	33.56	+1.2	**0.016**
	C	14.25	14.50	+1.8	**<0.001**	71.99	63.51	−11.8	0.982
	D	12.52	16.75	+33.7	**<0.001**	32.41	37.38	+15.3	**<0.001**
**Carbapenems**	Total	30.15	37.96	+25.9	**<0.001**	123.99	139.00	+12.1	**<0.001**
	A	14.72	6.39	−56.6	**<0.001**	163.53	120.45	−26.3	**<0.001**
	B	19.68	26.77	+36.0	**<0.001**	95.24	127.94	+34.3	**<0.001**
	C	42.16	58.30	+38.3	**<0.001**	195.83	182.31	−6.9	**<0.001**
	D	32.42	36.39	+12.2	**<0.001**	77.99	111.57	+43.1	**<0.001**

Values for antibiotic consumption are presented as DDDs/1000 patient-days. Data are compared using the Pearson’s chi-square test or Fisher’s exact test. ICU, intensive care unit.

**Table 2 antibiotics-11-00535-t002:** Prevalence of multidrug-resistant bacteria isolates from clinical samples.

		Ward				ICU			
Organism	Hospital	March 2018 –September 2019	March 2020 –September 2021	% Change	*p*	March 2018 –September 2019	March 2020 –September 2021	% Change	*p*
**MRSA**	Total	0.86	0.90	+4.7	**<0.001**	6.27	4.30	−31.4	**<0.001**
	A	1.22	1.70	+39.4	**<0.001**	10.43	11.79	+13.1	**0.002**
	B	0.84	0.86	+2.0	**0.013**	4.71	3.41	−27.6	0.077
	C	0.81	0.74	−8.3	0.396	6.54	3.76	−42.6	**<0.001**
	D	0.79	0.87	+11.1	**0.002**	6.01	4.25	−29.4	**0.015**
**VRE**	Total	0.46	0.69	+49.0	**<0.001**	1.51	1.91	+26.7	**<0.001**
	A	0.26	0.04	−85.2	**0.009**	2.14	2.55	+19.0	0.139
	B	0.27	0.37	+37.0	**0.001**	0.73	1.41	+92.5	**<0.001**
	C	0.50	0.59	+18.0	**0.002**	1.81	1.54	−14.8	0.625
	D	0.62	1.11	+80.0	**<0.001**	1.78	2.58	+45.5	**0.001**
**CRE**	Total	0.23	0.28	+22.4	**<0.001**	1.03	1.40	+36.4	**<0.001**
	A	0.12	0.13	+10.7	0.135	1.01	3.65	+259.9	**<0.001**
	B	0.19	0.23	+21.0	**0.035**	0.73	1.11	+52.0	**0.015**
	C	0.11	0.15	+39.2	**0.026**	0.30	0.50	+65.1	0.115
	D	0.39	0.45	+13.5	**0.007**	1.98	2.09	+5.5	0.154
**CRAB**	Total	0.79	0.74	−6.2	0.132	8.94	7.28	−18.6	**0.003**
	A	0.78	0.97	+24.6	**<0.001**	16.03	18.10	+13.0	**<0.001**
	B	0.45	0.62	+38.5	**<0.001**	6.33	7.70	+21.7	**<0.001**
	C	0.58	0.73	+25.8	**<0.001**	7.15	5.25	−26.5	**0.008**
	D	1.15	0.78	−32.7	**<0.001**	10.32	6.81	−34.0	**<0.001**
**CRPA**	Total	0.41	0.49	+20.1	**<0.001**	2.95	2.20	−25.7	**0.005**
	A	0.66	0.36	−46.1	0.536	8.58	9.23	+7.6	**0.022**
	B	0.52	0.57	+10.2	**0.008**	2.49	2.94	+18.0	**0.020**
	C	0.41	0.49	+18.8	**0.004**	2.58	1.19	−54.0	**<0.001**
	D	0.26	0.47	+79.9	**<0.001**	1.57	1.17	−25.6	0.342

Values for prevalence of MDR bacteria are presented as infection cases/1000 patient-days. Data are compared using the Mann–Whitney test. ICU, intensive care unit; MRSA, methicillin-resistant *Staphylococcus aureus*; VRE, vancomycin-resistant *Enterococcus*; CRE, carbapenem-resistant *Enterobacteriaceae*; CRAB, carbapenem-resistant *Acinetobacter baumannii*; CRPA, carbapenem-resistant *Pseudomonas aeruginosa*.

**Table 3 antibiotics-11-00535-t003:** Prevalence of multidrug-resistant bacteria isolates from surveillance samples.

		Ward				ICU			
Organism	Hospital	March 2018 –September 2019	March 2020 –September 2021	% Change	*p*	March 2018 –September 2019	March 2020 –September 2021	% Change	*p*
**MRSA**	Total	0.04	0.04	−14.7	0.921	6.34	3.87	−38.9	**<0.001**
	A	0.01	0.03	+77.1	0.274	2.68	1.46	−45.6	0.218
	B	0.02	0.01	−33.9	1.000	9.85	7.80	−20.8	0.157
	C	0.08	0.06	−29.9	0.546	2.34	0.57	−75.6	**<0.001**
	D	0.04	0.04	+2.9	0.662	7.56	3.87	−48.9	**<0.001**
**VRE**	Total	0.46	0.42	−8.7	0.475	3.76	1.31	−65.2	**<0.001**
	A	0.25	0.17	−34.2	0.829	1.97	1.94	−1.1	0.482
	B	0.92	0.47	−48.9	**<0.001**	9.14	1.60	−82.4	**<0.001**
	C	0.64	0.88	+37.7	**<0.001**	1.98	2.07	+4.5	0.445
	D	0.10	0.08	−21.6	0.634	0.55	0.20	−63.0	**0.021**
**CRE**	Total	0.52	0.73	+38.7	**<0.001**	2.20	1.97	−10.6	0.083
	A	0.19	0.44	+131.6	**<0.001**	3.28	5.47	+66.9	**<0.001**
	B	0.37	0.55	+49.2	**<0.001**	3.84	5.21	+35.7	**<0.001**
	C	0.23	0.31	+38.2	**0.002**	1.54	1.78	+15.8	0.201
	D	0.51	0.70	+36.1	**<0.001**	4.99	3.55	−28.8	**0.033**
**CRAB**	Total	0.01	0.01	−40.5	0.319	3.13	2.36	−24.6	**0.012**
	A	0.00	0.01	+43.0	0.055	0.59	0.09	−85.3	**0.001**
	B	0	0.01		0.075	4.68	6.27	+33.9	**<0.001**
	C	0.02	0.00	−77.6	0.248	6.76	5.73	−15.3	0.304
	D	0.04	0.01	−70.6	**0.040**	5.76	2.11	−63.3	**<0.001**

Values for prevalence of MDR bacteria are presented as infection cases/1000 patient-days. Data are compared using the Mann–Whitney test. ICU, intensive care unit; MRSA, methicillin-resistant *Staphylococcus aureus;* VRE, vancomycin-resistant *Enterococcus*; CRE, carbapenem-resistant *Enterobacteriaceae;* CRAB, carbapenem-resistant *Acinetobacter baumannii*.

## Data Availability

The dataset analyzed in this study can be found in [HARVARD Dataverse] [https://doi.org/10.7910/DVN/UIM9UL] (accessed on 11 April 2022).

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
