# Peer review of "Impact of COVID-19 on Antimicrobial Consumption and Spread of Multidrug-Resistance in Bacterial Infections"

_antibiotics, 2022, doi:10.3390/antibiotics11040535_

Round 1

Reviewer 1 Report

The article “Impact of the COVID-19 Pandemic on Antimicrobial Consumption and the Prevalence of Infections Caused by Multidrug-resistant Bacteria” of Kibum Jeon, Seri Jeong, Nuri Lee, Min-Jeong Park, Wonkeun Song, Han-Sung Kim, Hyun Soo Kim, and Jae-Seok Kim analyzes the differences in the consumption of antibiotics that are easily susceptible to resistance, and the prevalence of MDR-bacteria during the COVID-19 pandemic compared to the pre-pandemic period. The study was carried out on a wide sample of data on the use of antibiotics, which were collected from four hospitals in South Korea. The article is of interest as an independent scientific study, and also has practical significance for the further use in South Korea of those groups of antibiotics that the study concerned. The submitted manuscript is quite suitable for a special issue «Antibiotic Resistance in Asia - a One Health Perspective» by subject and content.

Some specific comments.

Lines 66. “The differences in antibiotic use and the prevalence of MDR bacteria before and dur- 66 ing the COVID-19 pandemic have been scarcely reported» - In my opinion, this is not quite true. There are a sufficient number of reviews on this topic, some of which the Authors refer to in the article. Perhaps the Authors should refer to this review (and other reviews too) in the Introduction.

Lines 78-81. Units of measurement should be specified for DDDs.

There are two figures with number 2 in the article, the first of them is really Figure 1.

The Tables lack an indication of the units of measurement of values that need to be specified either in the Title of each table, or in the corresponding columns, or in footnotes.

In the article there are 5 Tables with the results, perhaps some of the Tables can be given not in the main text of the article, but in Supplementary materials.

Discussion.

In the Discussion, references to the Tables and Figures should be given.

Lines 258-260. Aspect (iii). “To the best of our knowledge, this is the first study to stratify clinical and surveillance samples for the changes in the prevalence of MDR bacteria during COVID-19.» - There are some doubts. Perhaps this statement is true for South Korea.

The article may be accepted for publication after minor revisions.

Reviewer 2 Report

The presentation of results in the manuscript is excellent, and this is a good example of the effects of Covid-19 on the use of antibiotics and the prevalence of MDR.

All standard methods were used for the experiments and data collection. I found the document interesting for the readers and follow the scope of the journal Antibiotics.

I would recommend the article could be published in Antibiotics, after a minor revision. There are technical errors, I hope the editor will take care of them.

The authors need to address the below-mentioned queries.

  1. 1st paragraph (193-206) of the discussion could be moved to the introduction.
  2. The author could have discussed a bit rationale behind the prevalence of MRSA, VRE, CRE, and CRPA in the ward and VRE and CRE in the ICU from clinical samples increased and for surveillance samples, the prevalence of CRE in the ward increased.

  1. The author could have highlighted the reasons for the COVID-19 pandemic is associated with increased consumption of antibiotics and has influenced the prevalence of infections caused by MDR isolates in the discussion section.

  1. The author needs to update current data if available for covid cases (Lines 39-43).

  1. For Table 1, footnotes are missing.

  1. There is no figure 1 (both figures are Figure 2), and the author needs to increase the resolution of both figures.

  1. The author has discussed the results of consumption of antibiotics and the prevalence of MDR in an average of 4 hospitals, however, result from the individual hospital settings are different (some have positive and some have negative changes). The author needs to discuss the reason behind these differences.

  1. The author needs to comment on the large consumption of antibiotic carbapenem consumption in both locations.

  1. The author needs to discuss each entry of tables in the discussion.

  1. The author could include the following relevant references.

  • Guisado-Gil AB, Infante-Domínguez C, Peñalva G, Praena J, Roca C, Navarro-Amuedo MD, Aguilar-Guisado M, Espinosa-Aguilera N, Poyato-Borrego M, Romero-Rodríguez N, Aldabó T, Salto-Alejandre S, Ruiz-Pérez de Pipaón M, Lepe JA, Martín-Gutiérrez G, Gil-Navarro MV, Molina J, Pachón J, Cisneros JM, On Behalf Of The Prioam Team. Impact of the COVID-19 Pandemic on Antimicrobial Consumption and Hospital-Acquired Candidemia and Multidrug-Resistant Bloodstream Infections. Antibiotics (Basel). 2020 Nov 17;9(11):816. doi: 10.3390/antibiotics9110816. PMID: 33212785; PMCID: PMC7698104.
  • (b) Ruiz-Garbajosa, P., & Cantón, R. (2021). COVID-19: Impact on prescribing and antimicrobial resistance. Revista espanola de quimioterapia : publicacion oficial de la Sociedad Espanola de Quimioterapia34 Suppl 1(Suppl1), 63–68. https://doi.org/10.37201/req/s01.19.2021
  • (c) Ansari, Shamshul et al. “The potential impact of the COVID-19 pandemic on global antimicrobial and biocide resistance: an AMR Insights global perspective.” JAC-antimicrobial resistance 3,2 dlab038. 8 Apr. 2021, doi:10.1093/jacamr/dlab038

Reviewer 3 Report

Wonderful study has been conducted. There are very minor observations for improvement in document.

Title: Need modification as " Impact of COVID-19 on antimicrobial consumption and spread of multi drug resistance in bacterial infections"

Abstract: check for typos

Keywords: Revise keywords e.g last three keywords reflect same information while there is no word that can tell about prevalence

Introduction: Some typos exist

Results: Please mentioned below each table, what kind of stat was applied

Some of figures are not clear, this must be because of pasting in word file. So it is suggested to provide high quality pics with standard format.

Discussion:  It would be better if discussion is subdivided into subheadings as is done in results section.

Materials and Methods: Subsection 4.3 is not carrying any reference

Conclusion: Half of paragraph is in present while other half is in past form of English language. Please revise it

Query: Why authors have not studied vancomycin resistant Staphylococcus aureus (VRSA) and Vancomycin intermediate Staphylococcus aureus VISA) despite of higher consumption of vancomycin. This is just query while addressing this comment in article is not necessary
